# Lead Levels in the Most Consumed Mexican Foods: First Monitoring Effort

**DOI:** 10.3390/toxics12050318

**Published:** 2024-04-28

**Authors:** Alejandra Cantoral, Larissa Betanzos-Robledo, Sonia Collado-López, Betzabeth A. García-Martínez, Héctor Lamadrid-Figueroa, Rosa M. Mariscal-Moreno, Araceli Díaz-Ruiz, Camilo Ríos, Martha María Téllez-Rojo

**Affiliations:** 1Health Department, Iberoamericana University, Mexico City 01376, Mexico; alejandra.cantoral@ibero.mx (A.C.); rosa.mariscal@ibero.mx (R.M.M.-M.); 2Center for Nutrition and Health Research, National Institute of Public Health, Cuernavaca 62100, Mexico; mmtellez@insp.mx; 3Service of Basic Neuroscience, National Institute of Rehabilitation, Mexico City 14389, Mexico; bgarcia@correo.xoc.uam.mx; 4Department of Perinatal Health, Center for Population Health Research, National Institute of Public Health, Cuernavaca 62100, Mexico; hlamadrid@insp.mx; 5Department of Neurochemistry, National Institute of Neurology and Neurosurgery Manuel Velasco Suárez, Mexico City 14269, Mexico; adiaz@innn.edu.mx; 6Research Direction, National Institute of Rehabilitation, Mexico City 14389, Mexico; camrios@yahoo.com.mx

**Keywords:** lead, food safety, Mexico, baby foods, maximum limits

## Abstract

Globally, there is growing concern over the presence of lead (Pb) in foods because it is a heavy metal with several toxic effects on human health. However, monitoring studies have not been conducted in Mexico. In this study, we estimated the concentrations of Pb in the most consumed foods and identified those that exceeded the maximum limits (MLs) for Pb in foods established by the International Standards. Based on the Mexican National Health and Nutrition Survey, 103 foods and beverages were selected and purchased in Mexico City retail stores and markets. Samples were analyzed twice using atomic absorption spectrophotometry. Values above the limit of quantification (0.0025 mg/kg) were considered to be detected. The percentage of detected values was 18%. The highest concentration was found in infant rice cereal (1.005 mg/kg), whole wheat bread (0.447 mg/kg), pre-cooked rice (0.276 mg/kg), black pepper (0.239 mg/kg), and turmeric (0.176 mg/kg). Among the foods with detected Pb, the levels in infant rice cereal, whole wheat bread, pre-cooked rice, and soy infant formula exceeded the MLs. The food groups with the highest percentages of exceeded MLs were baby foods (18%) and cereals (11%). Monitoring the concentration of contaminants in foods is essential for implementing food safety policies and protecting consumer health.

## 1. Introduction

Lead (Pb) is considered one of the major public health concerns worldwide for food safety and security due to its severe detrimental effects on human health and the environment [1]. According to the World Health Organization (WHO), there is no level of Pb in the blood that can be considered risk-free, as this metal is toxic even at low concentrations, affecting the function of every human organ and system in which it is deposited [2]. 

The ingestion of food contaminated with Pb represents one of the most important human exposure routes for this toxic element. Pb, like other heavy metals, cannot decompose and is non-degradable, so it can accumulate along the food chain, becoming a threat to human health [3]. Its presence in food varies according to the different routes or sources of contamination, defined as intrinsic and extrinsic factors. Intrinsic factors include the seasons, soil, water, atmospheric deposits, and animal feeding regimens [4], as well as volcanic and vehicle emissions [5]. Extrinsic factors that contribute to food contamination include food technological processes, packaging, transportation, storage [4], culinary procedure tools, and cooking methods [6,7]. Once Pb is consumed through contaminated food, children can absorb 40–50% of the Pb oral dose, compared to 3–10% for adults [2]. In this sense, infants and children are considered the most vulnerable populations, particularly as Pb is a neurotoxicant that affects the developing central nervous system [8,9,10].

The prevention and/or reduction of exposure to Pb from foods includes recommended actions to avoid contamination of food along the food chain through good controls and practices during production, processing, distribution, and commercialization [11]. A key area by which to reduce Pb exposure worldwide is food safety and food regulations, through the monitoring of toxic heavy metal levels such as those of Pb in food [12]. However, these types of monitoring studies and programs have not been conducted in many countries like Mexico. 

Therefore, the aim of this study was to determine the concentrations of Pb in the foods, beverages, and spices most consumed by the Mexican population. As a secondary aim, we compared the Pb concentrations obtained with the maximum limits (MLs) established by the Codex Alimentarius Commission Food and Agriculture Organization of the United Nations and the World Health Organization (FAO/WHO) for Pb in foods [13], and with the MLs for candy according to the US Food and Drug Administration (FDA) [14], identifying foods that exceed these reference limits.

## 2. Materials and Methods

### 2.1. Sample Collection 

The foods and beverages most consumed by the Mexican population were identified using data from the 24 h recall applied in the National Health and Nutrition Survey [15], complemented with additional items previously identified as sources of Pb by our research group [16]. 

We identified retail outlets in Mexico City that represent the most probable options for food shopping and prioritized those that were easily accessible for our collectors to efficiently purchase all the food samples. Mexico City was selected as the site for food collection due to its prominence as the main and most densely populated city in Mexico [17]. From 12 April 2022 to 30 January 2023, a total of 103 food, beverage, and spice samples were collected from retail outlets (i.e., markets, supermarkets, and pharmacies) located in three counties: Iztapalapa, Benito Juárez, and Álvaro Obregón. Perishable foods such as fruits, vegetables, and meats were purchased in the largest markets in the city (“Central de Abastos” and “La Viga”); those markets are the main supply points for local markets. Processed foods were bought at the principal supermarket chains (i.e., Walmart, Bodega Aurrera, and San Pablo drug store). Detailed information about food items can be found in Appendix A.

### 2.2. Chemicals and Reagents

Standard solutions for Pb atomic absorption spectroscopy (1000 µg/mL) were used as certified calibration standards (Perkin Elmer, Norwalk, CT, USA). Bovine Liver Standard NIST 1577c (Sigma-Aldrich, St. Louis, MO, USA) was used as the internal control. Nitric acid (HNO_3_) 65% Suprapur^®^ (Merck, Darmstadt, Germany) was used to prepare acid digestions and calibration curves. Dibasic ammonium phosphate (Sigma-Aldrich, St. Louis, MO, USA) and Triton X-100 (Sigma-Aldrich, St. Louis, MO, USA) were used to prepare the matrix modifier. All solutions were prepared with deionized water obtained from a Direct-Q 3 UV purification system (Millipore, Bedford, MA, USA).

### 2.3. Sample Analysis and Quality Control

Samples were transported to the Neurochemistry Laboratory at the National Institute of Neurology and Neurosurgery in Mexico City for analysis. The system conditions were established following the recommendations of the equipment manual and digestion was performed following the technique proposed by Sifou [18]. All samples were unpacked and inspected, and samples that could decompose were stored at −20 °C until further experimentation to avoid decomposition. Foods packaged in paper or cardboard, plastic, or other containers were only cleaned with deionized water twice. Fruits were washed with soap and deionized water (twice). Those with inedible peel (such as lemon, mango, and orange) were peeled, and only the pulp was used for the next steps. In the case of fruits like apples (commonly consumed with peel) the whole fruit was used. Meats (chicken, beef, pork) and eggs (without shell) were not cooked and were sampled raw for the next steps. Cereals and legumes were analyzed without processing (raw samples). Solid items underwent dehydration at 80 °C for 72 h and liquid samples were processed in wet weight. All solid items were ground using a house grinder and finally stored in polypropylene tubes until analysis. The liquid and solid samples were weighed in duplicate (0.1–0.2 g for solids or 1 mL for liquids) in test tubes, and 2 mL of 65% Suprapur^®^ HNO_3_ was added. The tubes were covered, mixed, and left at room temperature for 12 h. Subsequently, they were placed in a water bath (Labline instrument, shaking water bath) at 60 °C until a clear solution was obtained, and samples of 100 µL were taken for analysis. 

During the development of the system validation tests, which were carried out following the recommendations established by the Commission for Analytical Control and Expansion of Coverage (CCAYAC-CR-03/0), a repeatability test was performed. This test involves evaluating the lower limit of quantification by five-fold (2.5 µg/L, lower value of the calibration curve) and three concentration values located within the calibration curve: low level 8 µg/L, medium level 25 µg/L, and high level 40 µg/L. For compliance with this test, the values obtained should show a maximum coefficient of variation percentage of 20% for the limit of quantification and 15% for the rest of the determinations. The results obtained were 8.9%, 4.8%, 2.8%, and 0.8%, respectively. Therefore, the proposed method complied with the repeatability test. We considered compliance through validation of the method used to perform duplicate testing.

The Pb levels were determined using an atomic absorption spectrophotometer (AAS) (Perkin Elmer AAnalyst-600) equipped with a graphite furnace HGA-600 and coupled to an AS800 autosampler. Calibration curve solutions were prepared each day of the analysis was conducted by diluting a standard solution with 0.2% ultrapure HNO_3_; the coefficient of determination was at least 0.99. The metal content was calculated using a calibration curve in a concentration range 2.5 to 45 µg/L. A 1:1 dilution of the acid digestion was performed with a matrix modifier containing 0.2 mL of dibasic ammonium phosphate, 0.5 mL of Triton X-100, and 0.2 mL of ultrapure HNO_3_ in 100 mL of deionized water. A volume of 20 µL of the solution was injected into the equipment. Pb determination was assessed in duplicate for each sample. As an internal control, on every day of analysis, a solution of acid digestion of bovine liver standard (similar digestion as the samples) equivalent to 3.5 µg/L of Pb was analyzed every 30 samples. From the controls analyzed, a percentage recovery of 105.03 ± 9.01% was obtained. The lower limit of quantification (LoQ) was 0.0025 mg/kg; values >LoQ were considered to be detected. 

### 2.4. Statistical Analysis

Pb levels (mg/kg) were determined in duplicate in each sample and the average was obtained (mean and standard deviation [SD]). The samples were classified into 13 groups: (1) baby foods; (2) beverages; (3) candy and snacks; (4) cereals; (5) condiments and spices; (6) dairy products; (7) fats and oils; (8) fruits; (9) legumes; (10) meats, sausages, and eggs; (11) seafood; (12) soups; and (13) vegetables. Each food item was compared with its corresponding MLs established for Pb by the FAO/WHO Codex Alimentarius Commission, which are established only for some types of food groups such as vegetables, fruits, pulses, cereal grain, infant formula, fish, meat, and alcohol products [19]. Because of the lack of FAO/WHO MLs for the candy and snacks group, this group was compared with the US FDA’s recommended lead concentration in candy that is likely to be consumed frequently by small children of 0.1 mg/kg (limit set for producers) [14].

Finally, the percentage of detected Pb levels (above the LoQ) and the percentage of those that exceeded the MLs were estimated per food group. 

## 3. Results and Discussion 

The complete list of the 103 food and beverage samples evaluated is presented in Table 1. The detection of Pb (>LoQ) in the total sample was 18% (n = 19), with the detected levels ranging from 0.021 mg/kg to 1.005 mg/kg. The highest concentration of Pb was found in infant rice cereal (brand 2) (1.005 mg/kg), whole wheat bread (0.447 mg/kg), pre-cooked rice (0.276 mg/kg), black pepper (0.239 mg/kg), and turmeric (0.176 mg/kg). 

The food groups with more detected Pb levels were as follows: (1) condiments and spices with 45% (n = 5/11) of detected levels varying from 0.021 to 0.239 mg/kg; tea sachets had the lowest levels, and black pepper had the highest Pb levels in this food group; (2) meats, sausages, and eggs with 43% (n = 3/7) of detected levels varying from 0.026 mg/kg to 0.133 mg/kg corresponding to turkey sausages (brand 2) and beef liver, respectively; (3) cereals with 33% (n = 6/18) of detected levels, with 0.030 mg/kg being the lowest level identified in rice cake and 0.447 mg/kg being the highest level in whole wheat bread; and (4) baby foods with 18% (n = 2/11) of detected values corresponding to infant formula soy milk (brand 2) (0.035 mg/kg) and infant rice cereal (brand 2) (1.005 mg/kg). The food groups with Pb levels < LoQ were beverages, dairy products, seafood, fruits, fats and oils, soups, and vegetables.

Among the 19 food items with detected Pb levels in the present study, we identified 12 food items with FAO/WHO MLs established for Pb, and from these, only four food items exceeded these MLs. These items were infant rice cereal (brand 2), whole wheat bread, pre-cooked rice, and soy infant formula (brand 2). Finally, the food groups with the highest percentages of samples that exceeded the FAO/WHO MLs for Pb were baby foods (18%) and cereals (11%). 

In Mexico, there are official standards aimed at regulating the levels of heavy metals in food throughout the country. Among the foods with established Mexican MLs for Pb are infant formulas (0.02 mg/kg), juices, nectars (0.3 mg/kg), vegetables (1.0 mg/kg), meat products (1.0 mg/kg), dairy products (0.2 mg/kg), fish products (ranging from 0.5 to 2.0 mg/kg), and flour (0.5 mg/kg) [20,21,22]. In general, the concentrations obtained in the present study were found to be below Mexican regulation levels, except for soy infant formula with a Pb level of 0.035 mg/kg. However, the Mexican limits are above those standards established internationally as the FAO/WHO MLs. Additionally, no limits were updated in the Mexican regulations for foods such as cereals and fruits.

This is the first analysis that reports the Pb levels in the most common foods consumed by the Mexican population. Eighteen percent of the food items analyzed had detectable Pb levels. Products whose main ingredients are from rice and wheat presented the highest concentrations of Pb (infant rice cereal (1.005 mg/kg), whole wheat bread (0.447 mg/kg), pre-cooked rice (0.276 mg/kg)), and these same items are three of the four items that exceeded the FAO/OMS MLs for Pb (0.2 mg/kg in cereals).

Our results are consistent with a previous review performed by our group, where we summarized the existing evidence on heavy metal exposure in natural or minimally processed foods for human consumption worldwide that included 152 articles. We found that both cereals (rice and wheat) were reported in the literature as sources of Pb, especially in Iran and China, where the Pb-reported levels for wheat exceed the FAO/OMS MLs [16]. Another study in China, which encompassed 1386 articles (n = 391,633 samples) published between 2010 and 2020 in the country, evaluated concentrations of Pb in the edible parts of grains, vegetables, meat, eggs, milk, and fruit, showing that Pb in foods ranged from 0.09 mg/kg to 0.30 mg/kg, and 87% of the Pb intake was contributed by grains, vegetables, aquatic products, mushrooms, and meat [23]. This study agreed with ours, showing that grains represent one of the food groups with major concentrations of Pb.

Other studies have reported concentrations of Pb in rice. A study in Spain that determined the concentrations of toxic metals in cooked and digested rice in a total of 42 samples (evaluated 14 varieties by triplicate) found Pb mean levels (SD) of 0.14 (0.15) mg/kg and 0.06 (0.04) mg/kg in cooked and digested rice, respectively [24]. Another study determined Pb levels in white rice and brown rice grown in the US and other countries (Thailand, India, and Italy) and found that Pb mean levels were 0.0056 mg/kg (range 0.0002 mg/kg to 0.032 mg/kg) in US white rice, 0.0074 mg/kg (0.0014 mg/kg to 0.034 mg/kg) in US brown rice, and 0.014 mg/kg (range 0.002 mg/kg to 0.096 mg/kg) in rice from other countries [25]. However, in contrast with our results, none of the samples of both studies exceeded the FAO/OMS MLs set for Pb (0.2 mg/kg in cereals).

It was identified that a relevant source of Pb for rice and wheat crops is the atmospheric deposition of Pb that can be transferred from leaves to grain [26]. Particularly for wheat, the contribution rates of atmospheric deposition in wheat roots, stems, leaves, and grains were 14%, 66%, 84%, and 77%, respectively [27]. In the case of rice, a high contribution of atmospheric exposure to Pb levels in rice grain was also evaluated [26]. Notably, in our analysis, the samples of rice in their natural form (n = 3) had Pb levels < LoQ, but the items with Pb levels detected in the cereal groups (pre-cooked rice, rice cake, sweet bread, whole wheat bread, and wheat flour) were rice- and wheat-based products. Additionally, with respect to atmospheric deposition, in our results, an industrial process could be involved in the Pb levels detected in these products.

In this study, infant rice cereal and soy infant formula were among the food items with the highest Pb levels; these baby foods had the highest percentages exceeding FAO/WHO MLs (18%). This result is not surprising, as a previous study by Healthy Babies Bright Futures (HBBF), in which 168 baby foods were analyzed, identified that Pb was present in 94% of the samples; the content varied from 0.0031 to 0.0674 mg/kg in different infant rice cereal brands and was 0.0078 mg/kg for soy infant formula [28]. In a systematic review that evaluated Pb levels in foods consumed or produced in Brazil, a total of 77 articles from 8466 food samples were found to have Pb levels ranging from 0.0004 mg/kg for nuts and cocoa products to 0.4842 mg/kg for infant food; this is consistent with baby foods having the highest Pb levels [29]. The presence of heavy metals in baby and infant foods has become so relevant that the US FDA agency has implemented a science-based strategy called “Closer to Zero”. This initiative aims at reducing exposure to heavy metals to the lowest possible levels in this kind of food [30]. Recently, as part of the plan, the FDA updated the proposed levels for Pb in baby foods, intending to significantly reduce Pb exposure from food while ensuring the availability of nutritious food [31].

In our study, condiments and spices were the group with a higher percentage of Pb levels detected in 45% of the samples, ranging from 0.021 mg/kg to 0.239 mg/kg. However, a comparison with the FAO/WHO MLs could not be performed because the FAO/OMS has not established MLs for most of the food items in this group, such as turmeric, paprika, and black pepper. A US Consumer Report in 2021 analyzed 15 types of dried herbs and spices produced by national and private-label brands (n = 126) and identified that 40 tested products had elevated toxic elements levels, including Pb, highlighting that in 31 products, Pb levels were so high that they exceeded the maximum amount anyone should have in a day [32]. The adulteration of spices with Pb is a topic of particular concern; there is evidence that the consumption of turmeric contaminated with Pb is associated with childhood lead poisoning cases. Between 2010 and 2014, cases of childhood lead poisoning that were attributable to culinary spice consumption were reported in the US. Children’s Hospital Boston analyzed Indian spices from cases of pediatric lead poisoning, finding that 25% contained >1 mcg/g Pb, and concluded that chronic exposure to these spices can increase blood Pb levels [33,34]. 

The meats, sausages, and eggs group in our study was another group with a higher percentage of detected Pb levels (43% of samples) in the range of 0.026 mg/kg to 0.133 mg/kg. Although these levels were not above the MLs, similar findings were reported in Italy, where the detected Pb levels in white meat (0.003189 mg/kg) and processed meat (0.00981 mg/kg) were below the MLs [35], and also in Spain, where chicken, pork, and beef presented Pb values of 0.00694 mg/kg, 0.005 mg/kg, and 0.00191 mg/kg, respectively [36]. 

Finally, our results show that Pb was present in soy (natural and industrialized); the presence of Pb in isolated soy could be caused by contamination during processing as this has previously been documented [37]. Foods can be contaminated by Pb during different steps of production and processing. As mentioned previously, wheat and rice can obtain Pb from contaminated water and soil, and in the case of soy, Pb can be derived from the industrialized process. This reflects the multitude of factors that can modify the concentration of contaminants in food, highlighting the urgent need to update Mexican regulations and maintain continuous monitoring of food production. 

In contrast with our results, where dairy products and vegetables were among the food groups with non-detectable Pb levels, previous studies in Mexico demonstrated high Pb levels in these food groups. One study determined Pb levels in milk and cheese samples produced in an area irrigated with wastewater from Puebla state and found that the mean Pb levels detected in milk, milk whey, and ranchero cheese were 0.03 mg/kg, 0.07 mg/kg, and 0.11 mg/kg, respectively; all values were above the FAO/OMS MLs of 0.020 mg/kg for milk and cheese [38]. Another study detected Pb levels in milk from cows fed with alfalfa produced in soils irrigated with wastewater in Puebla and Tlaxcala; Pb levels were in the range of 0.039 to 0.059 mg/kg and were also above the international MLs [38]. Notably, these are among the Mexican states with the highest reported prevalence of Pb poisoning in children, 46.6% and 35.6%, respectively [39].

Another study carried out in Zacatecas state evaluated Pb levels in the edible parts of vegetables cultivated in agricultural soils contaminated by historical mining. The study classified vegetables into root vegetables (carrot and garlic) and fruit vegetables (pepper) finding Pb mean levels of 9.6 mg/kg and 4.8 mg/kg in fruit vegetables and root vegetables, respectively; these levels exceeded the MLs established by the FAO/WHO of 0.05 mg/kg (fruit vegetables) and 0.1 mg/kg (roots vegetables) [40].

A review that evaluated the heavy metal and metalloid pollution status in soil, water, and foods in Bangladesh reported elevated Pb concentrations in the vegetable food group, comprising food items such as spinach (11.48 mg/kg), cabbage (22.09 mg/kg), and carrots (0.006 mg/kg) [41]. In our study, all food items in the vegetable group were found to have Pb levels < LoQ.

It is important to note that the studies mentioned above were carried out in areas previously identified as contaminated. In our study, we could not determine whether food items were cultivated in a highly contaminated area. 

One of the key areas of focus in public health is food safety and food regulations. Total diet studies (TDSs) are one of the most cost-effective tools for assessing dietary exposure to elements such as Pb. Thirty-three countries have performed TDSs [12], such as the United States (US), which has conducted these studies since 1961 through the Food and Drug Administration (FDA) to monitor nutrients and contaminant levels in foods commonly consumed by the US population [13]. Additionally, international agencies such as the FAO/WHO have established standards for MLs for Pb and other toxic substances in food to promote the safety of food production worldwide [19]. However, there is evidence of the presence of heavy metals exceeding these limits throughout the world and in all food groups, even in uncontaminated areas. Therefore, it is important that monitoring be performed in each country and in key locations within the country, to assess compliance with national and international standards. However, these types of monitoring studies and programs have not been conducted in Mexico.

This first analysis of Pb levels in food is part of a risk communication action not only to increase the awareness of consumers, food producers, the food industry, and stakeholders but also to open a dialogue and suggest actions to promote the implementation of a food monitoring system that is able to identify the sources of Pb (intrinsic or extrinsic factors) and reduce them in the food chain. As an example, through the different Chinese TDSs performed since 1990, the Pb levels in food as the daily dietary intake decreased from 86.3 μg to 34.4 μg [42]. In addition, in 2010, the joint FAO/WHO Expert Committee on Food Additives (JECFA) established that no level of Pb intake is entirely safe [43] and suggested that all countries should reinforce efforts to reduce Pb in foods. However, the presence of a contaminant does not mean that food is unsafe for consumption. Although it is not possible to eliminate these elements completely from food, we expect that the recommended action levels will cause manufacturers to implement agricultural and processing measures to reduce Pb levels in their food products, reducing the potential adverse effects associated with exposure to Pb from food.

One of the limitations of this study is that all food samples were purchased in Mexico City; therefore, regional variations in the sources of exposure could not be detected. Another limitation is the small sample size of the food items analyzed in our study. Our results should be interpreted with caution and may not be representative of the food groups included in this sense, as very high values may be outliers. Despite the fact that we tried to include the most representative foods of the Mexican diet, it is important to consider conducting food monitoring studies on larger food samples given the high production of foodstuffs. Lastly, the present study is not a systematic food monitoring study in Mexico but rather an effort to call attention to the need for systematic monitoring, with replications with national representativeness and variability in the geographical distribution. 

In addition to recognizing that the main source of Pb exposure in the general Mexican population is the use of lead-glazed ceramic for cooking or serving food [44], considering all the evidence, particular attention must be focused on foods consumed since pregnancy, as prenatal blood Pb concentrations as low as 1–5 µg/dL may affect neurodevelopment in the developing fetus, especially language development [45]. In addition, during childhood, Pb accumulates in the body and can cause adverse and permanent neurodevelopmental consequences and other “silent consequences” [46]. This is especially relevant as Pb is accumulated in the body for decades, and in adult life, it affects the reproductive system [47] and cardiovascular system [48], among many others. In Mexico, the problem of Pb exposure is estimated to be one of the highest worldwide, as 17% of children (1–4 years of age) present blood Pb levels above 5 μg/dL [39].

## 4. Conclusions

In this study, we determined the concentrations of Pb in the foods, beverages, condiments, and spices most consumed by the Mexican population, and have contributed to the limited evidence about Pb’s presence in Mexican food. Mexico has a food monitoring system; however, the information is not made public or published regularly. Therefore, this report is an effort to demonstrate the importance and need for constant food monitoring concerning contamination with toxic elements, giving the population access to information from which to make informed decisions with respect to food consumption. 

## Figures and Tables

**Table 1 toxics-12-00318-t001:** Lead (Pb) levels in the most consumed Mexican foods (n = 103) per food group and comparison with the maximum levels (MLs) for Pb established by the FAO/WHO.

Food Groups and Food Items (n = 103)	Pb Mean (SD) Levels (mg/kg)	Detected Pb (%) by Food Group	FAO/WHO MLs ^a^ (mg/kg)	Exceeds MLs (%)
**Baby foods (n = 11)**				
Infant Formula Whole Milk (Brand 1)	<LoQ	18%	0.01	18%
Infant Formula Soy Milk (Brand 1)	<LoQ		0.01	
Infant Formula Whole Milk (Brand 2)	<LoQ		0.01	
**Infant Formula Soy Milk (Brand 2)**	**0.035 (0.008)**		**0.01**	
Apple Juice (Brand 1)	<LoQ		0.03 ^^^	
Carrot Puree	<LoQ		NI	
Chicken, Vegetables, and Rice Porridge	<LoQ		NI	
Strawberry and Apple Snack	<LoQ		0.20 ^^^	
**Rice Cereal (Brand 2)**	**1.005 (0.042)**		**0.20 ^^^**	
Infant Formula Whole Milk (Brand 3)	<LoQ		0.02	
Infant Grow and Gain Strawberry Shake	<LoQ		0.02	
**Beverages (n = 2)**				
Soluble Coffee (Brand 1)	<LoQ	0%	NI	0%
Bottled Soft Drink (Brand 1)	<LoQ		NI	
**Candy and Snacks ^b^ (n = 16)**				
Cacao Powder (Bulk)	0.083 (0.032)	13%	0.10	0%
Chocolate Bar (Brand 1)	<LoQ		0.10	
Chocolate Powder (Brand 1)	<LoQ		0.10	
Chamoy Candy (Brand 1)	<LoQ		0.10	
Chewing Gum (Brand 1)	<LoQ		0.10	
Popsicle (Brand 1)	<LoQ		0.10	
Marzipan Candy (Brand 1)	<LoQ		0.10	
Poblano Candy “borrachito” (Brand 1)	<LoQ		0.10	
Tamarind Poblano Candy (Brand 1)	0.050 (0.001)		0.10	
Tamarind Popsicle (Brand 1)	<LoQ		0.10	
Tamarind Candy (Brand 1)	<LoQ		0.10	
Wheat Chips (Brand 1)	<LoQ		0.20 *	
Sweet Cookie (Brand 1)	<LoQ		0.20 *	
Jelly (Brand 1)	<LoQ		0.10	
Potato Chips (Brand 1)	<LoQ		0.10 *	
Potato Chips (Brand 2)	<LoQ		0.10 *	
**Cereals (n = 18)**				
Rice Flour (Brand 1)	<LoQ	33%	0.20	11%
Rice “*Oryza Sativa*” (Brand 1)	<LoQ		0.20	
Rice “*Oryza Sativa*”(Brand 2)	<LoQ		0.20	
Rice “*Oryza Sativa*” (Brand 3)	<LoQ		0.20	
**Pre-cooked Rice (Brand 1)**	**0.276 (0.017)**		**0.20**	
Oat “*Avena Sativa*” (Brand 1)	<LoQ		0.20	
Breakfast Cereal (Brand 1)	<LoQ		0.20 ^^^	
Wheat Cookies (Brand 1)	<LoQ		0.20 ^^^	
Crackers (Brand 1)	<LoQ		0.20 ^^^	
Rice Cake (Brand 1)	0.030 (0.012)		0.20 ^^^	
Corn Flour (Brand 1)	<LoQ		NI	
White Bread (Brand 1)	<LoQ		NI	
Sweet Bread (Brand 1)	0.123 (0.020)		0.20 ^^^	
**Whole Wheat Bread (Brand 1)**	**0.447 (0.192)**		**0.20 ^^^**	
Breadcrumbs (Brand 1)	<LoQ		0.20 ^^^	
Wheat Flour (Brand 1)	0.031 (0.009)		0.20 ^^^	
Wheat Flour (Brand 2)	0.070 (0.007)		0.20 ^^^	
Wheat Tortillas (Brand 1)	<LoQ		0.20 ^^^	
**Condiments and spices (n = 11)**				
Chicken Broth Cubes (Brand 1)	<LoQ	45%	NI	0%
Chili Powder (Brand 1)	<LoQ		NI	
Ancho Chili “*Capsicum annuum*” (Brand 1)	<LoQ		0.05	
Guajillo Chili “*Capsicum annuum*” (Brand 1)	0.037 (0.000)		0.05	
Fresh Green Chili (Bulk) “*Capsicum annuum* ‘*serrano*’”	<LoQ		0.05	
Canned Green Chili (Brand 1)	<LoQ		0.10	
Turmeric “*Curcuma longa*” (Bulk)	0.176 (0.032)		NI	
Paprika “*Capsicum*” (Bulk)	0.092 (0.027)		NI	
Black Pepper “*Piper nigrum* L.” (Bulk)	0.239 (0.007)		NI	
Industrialized Sauce (Brand 1)	<LoQ		NI	
Tea Sachet (Brand 1)	0.021 (0.003)		NI	
**Dairy Products (n = 5)**				
Whole Liquid Milk (Brand 2)	<LoQ	0%	0.02	0%
Whole Liquid Milk (Brand 1)	<LoQ		0.02	
Asadero Cheese (Brand 1)	<LoQ		0.02	
Manchego Cheese (Brand 2)	<LoQ		0.02	
Natural Yogurt (Brand 1)	<LoQ		0.02	
**Fats and oils (n = 4)**				
Vegetable Oil (Brand 1)	<LoQ	0%	0.08	0%
Vegetable Oil (Brand 2)	<LoQ		0.08	
Sour Cream (Brand 2)	<LoQ		0.02	
Mayonnaise (Brand 1)	<LoQ		0.04	
**Fruits (n = 7)**				
Guava “*Psidium guajava*”	<LoQ	0%	0.10	0%
Lemon “*Citrus limon*”	<LoQ		0.10	
Ataulfo Mango “*Mangifera indica*”	<LoQ		0.10	
Golden Yellow Apple “*Malus domestica* ‘*golden delicious*’”	<LoQ		0.10	
Orange “*Citrus sinensis* L.”	<LoQ		0.10	
Grapefruit “*Citrus Paradisi*”	<LoQ		0.10	
Grape “*Vitis vinifera*”	<LoQ		0.10	
**Legumes (n = 6)**				
White Bean “*Phaseolus vulgaris*” (Brand 1)	<LoQ	17%	0.10	0%
Black Canned Beans (Brand 1)	<LoQ		0.10	
Black Beans “*Phaseolus vulgaris*” (Brand 1)	<LoQ		0.10	
Lentils “*Lens culinaris*” (Brand 1)	<LoQ		0.10	
Lentil Instant Soup (Brand 1)	<LoQ		0.10	
Soybean “*Glycine max l. Merr*.” (Brand 1)	0.029 (0.013)		0.10	
**Meats, sausages, and eggs (n = 7)**				
Beef Liver (Brand 1)	0.133 (0.015)	43%	0.20	0%
Egg (Brand 1)	<LoQ		0.05	
Egg (Brand 2)	<LoQ		0.05	
Pork Ham (Brand 1)	0.062 (0.003)		0.15 ^^^	
Chicken Breast (Bulk)	<LoQ		0.10	
Turkey Sausages (Brand 1)	<LoQ		0.10 ^^^	
Turkey sausages (Brand 2)	0.026 (0.010)		0.10 ^^^	
**Seafood (n = 3)**				
Canned Tuna (Brand 1)	<LoQ	0%	NI	0%
Canned Tuna (Brand 2)	<LoQ		NI	
Pacotilla Shrimp “*Litopenaeus vannamei*”	<LoQ		0.30 ^^^	
**Soups (n = 3)**				
Instant Pasta Soup (Brad 1)	<LoQ	0%	0.20 ^^^	0%
Canned Vegetable Soup (Brand 1)	<LoQ		0.02	
Pasta Soup to Prepare (Brand 1)	<LoQ		0.20^^^	
**Vegetables (n = 10)**				
White Onion “*Allium cepa*”	<LoQ	0%	0.10	0%
Mushrooms “*Agaricus bisporus*”	<LoQ		0.30	
Chayote “*Sechium edule*”	<LoQ		0.05	
Coriander “*Coriandrum sativum* “	<LoQ		0.30	
Cabbage “*Brassica oleracea car. capitata*”	<LoQ		0.10	
Corn “*Zea mays*”	<LoQ		0.05	
Chard “*Beta vulgaris var. cicla*”	<LoQ		0.30	
Spinach “*Spinacia oleracea*”	<LoQ		NI	
Nopal Cactus “*Opuntia ficus-indica* (L.)”	<LoQ		0.30	
Carrot “*Daucus carota* subsp. *sativus*”	<LoQ		0.10	

Abbreviations: Pb: lead; MLs: maximum levels; <LoQ: below the limit of quantification (<0.0025 mg/kg); NI; no information. ^a^ Comparisons were performed using the FAO and WHO Maximum Levels for Lead, General Standard for Contaminants and Toxins in Food and Feed. ^b^ Comparisons were performed using the US FDA Recommended Lead Concentration in Candy likely to be Consumed Frequently by small Children (0.1 mg/kg). * These food items were categorized as candy and snacks but were compared with FAO/WHO MLs because of their nature (cereal grains; root and tuber vegetables). ^ These foods were grouped in this food group by their nature, although FAO/WHO MLs do not specify these foods.

## Data Availability

The data presented in this study are available on request from the corresponding author.

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
