# Peer review of "Lead Levels in the Most Consumed Mexican Foods: First Monitoring Effort"

_toxics, 2024, doi:10.3390/toxics12050318_

Round 1
Reviewer 1 Report
Comments and Suggestions for Authors
This manuscript is well written, the quality of presentation is good enough and the introduction is focused on the purpose but, in its current form, is similar to a quality control laboratory analytical report rather than a scientific article since there is no relevant innovation on the applied methodology or a significant original approach.
The general picture of the health risk emerged from this work is relatively positive since “only” 4 out of 103 samples had a lead concentration higher than the ML. Three of the four samples that exceeded the ML belong to cereal samples (one baby food sample and two samples from the cereal group). This trouble is well known as reported in literature (29,32,34). In my opinion, it would be useful to focus the investigations by expanding the number of these samples by collecting them from different production areas looking for any correlation between contamination and geographical origin.
The results obtained on the 11 samples belonging to “baby food” do not agree with the aforementioned HBBF research (reference 38) where 168 samples were analyzed and the lead concentration ranged between 3 and 67 ppb. In this context, the “rice cereal (Brand2)” sample (1034 ppb lead) appears to be an outlier; the number of samples should also be expanded for this group.
These limits were recognized by the authors themselves (lines 266-270).
Therefore, the experimental design should be improved.
Additional comments:Table 1
1) <LoQ in the SD column is no sense
2) Pork Ham(brand1): the reported SD is wrong likely due typing mistake
Author Response
Reviewer 1
This manuscript is well written, the quality of presentation is good enough and the introduction is focused on the purpose but, in its current form, is similar to a quality control laboratory analytical report rather than a scientific article since there is no relevant innovation on the applied methodology or a significant original approach.
The general picture of the health risk emerged from this work is relatively positive since “only” 4 out of 103 samples had a lead concentration higher than the ML. Three of the four samples that exceeded the ML belong to cereal samples (one baby food sample and two samples from the cereal group). This trouble is well known as reported in literature (29,32,34). In my opinion, it would be useful to focus the investigations by expanding the number of these samples by collecting them from different production areas looking for any correlation between contamination and geographical origin.
The results obtained on the 11 samples belonging to “baby food” do not agree with the aforementioned HBBF research (reference 38) where 168 samples were analyzed and the lead concentration ranged between 3 and 67 ppb. In this context, the “rice cereal (Brand2)” sample (1034 ppb lead) appears to be an outlier; the number of samples should also be expanded for this group.
These limits were recognized by the authors themselves (lines 266-270).
Therefore, the experimental design should be improved.
R: Thanks to the reviewer for the insightful comments. They improve a lot our manuscript.
Even the authors recognize that this is not an innovative methodology. The importance of the “monitoring study” emerges from the local context. Mexico has been dealing with lead exposure for centuries. Even though this problem is antique, there is no effort to monitor food as other countries have done through total diet studies.
Regarding the Pb level detected in baby rice cereal (Brand 2), we corroborated this value, which is correct. Additionally, we recognize the small sample size as a limitation, and some of our lead concentration results differ from those of other studies. Still, we highlight that this is the first monitoring study effort in Mexico, and we aimed to have a starting point to estimate the magnitude and distribution of this problem (lines 342-350). Also, we unified all reported units along the text in mg/kg to be consistent with our results.
Additional comments:Table 1
1) <LoQ in the SD column is no sense
R: Thanks for this observation. We modified the table, deleted the SD column, and included the SD value only in the items with a mean Pb level >LoQ.
2) Pork Ham(brand1): the reported SD is wrong likely due typing mistake
R: Thanks for this observation. We have verified the information, and this data is correct.
Reviewer 2 Report
Comments and Suggestions for Authors
The introduction section needs a lot of punctuation check. Lack and unnecessary use of punctuation marks.
Still, in the introduction, there is no logical flow! Each succeeding sentence and paragraphs have no chronological link to the preceding one. The introduction needs complete overhauling.
Authors should submit this article for professional proofreading to enhance the lexis and structure of the English language in this work.
Method section:
In subsection 2.1 "Sample collection and preparation" the authors never mentioned how collected samples were prepared. The authors might want to rename this subsection as "sample collection."
The manufacturers/suppliers, city, and country of where chemicals that were used (HNO3, Triton X-100, dibasic ammonium phosphate) need to be provided. Additionally, information about the lead standard used should also be provided.
Similarly, the make and manufacturer of the instrument supplying the used deionized water should be provided. Also, the make and type of the water bath used should be provided.
Generally, appropriate information on the chemicals and instruments used should be provided.
Subsection 2.3
Why did the authors conduct a duplicate analysis and not a triplicate analysis? Statistically, triplicate analysis is more robust than duplicate analysis. The authors should justify their reason.
In results section
Table 1 is not presented in a standard form suitable for journal publication. A 3-wier table format is more appropriate.
The table is too long, boring and difficult to follow. The authors should decongest and summarize the table for easy readability.
In discussion section
The sentence in lines 186-188 is difficult to comprehend. This should be re-written.
I would suggest that sections 3 (results) and section 4 (discussion) should be merged as one section (results and discussion). This would make it easy for readers to link the concentration of Pb in this study to an explanation and comparison the authors intend with other studies. In this current form, it is difficult to go back and forth between the two sections to comprehend the authors' message, especially in the discussion section.
Again, the authors need to provide a concentration of any food product and not just its percentage contribution as it is in its current form. Providing the concentration detected in your discussion in comparison with other studies or MLs by FAO/WHO would give the readers an idea of the level of Pb contamination in that food sample.
The authors need to be consistent with the use of scientific units. The author should stick to either concentration unit in ppb or mg/kg. Moreso, mg/kg is equivalent to concentration in ppm and not ppb. The authors need to be consistent. In lines 218-223, the authors referenced a study and reported in ppb.
The results were not critically and adequately discussed. Most of the results in this study were limited to other studies within Mexico. Putting it in the perspective of the global reader, the authors should make a more robust comparison with other studies within North/Latin America, and other continents. This would give the article a wider reach of audience, and room for global comparison.
Most of the information in the latter part of the discussion is supposed to be in the introduction section. Also, there are too many repetitions of things already mentioned in the introduction section that are repeated in the discussion.
Conclusion section
The conclusion section did not mention anything about the results of this study and did not conclude on anything either.
The conclusion was poorly written. It needs to be completely rewritten.
References
A full bibliography of reference no 1 should be provided.
Comments on the Quality of English LanguageThe quality of English in this work is poor. This article could use professional proofreading.
Author Response
Reviewer 2
- The introduction section needs a lot of punctuation check. Lack and unnecessary use of punctuation marks.
R: Thanks for the reviewers' insightful comments. They improve our paper a lot. We re-wrote the introduction section, and we will pay for the editing service of MDPI.
- Still, in the introduction, there is no logical flow! Each succeeding sentence and paragraphs have no chronological link to the preceding one. The introduction needs complete overhauling.
R: Thanks for this observation; we re-wrote the introduction section.
- Authors should submit this article for professional proofreading to enhance the lexis and structure of the English language in this work.
R: Thanks for this observation, we will pay for the editing service of MDPI
Method section:
- In subsection 2.1 "Sample collection and preparation" the authors never mentioned how collected samples were prepared. The authors might want to rename this subsection as "sample collection."
R: We appreciate your feedback. In line 76, the section was renamed as you suggested: 2.1 “Sample collection”.
- The manufacturers/suppliers, city, and country of where chemicals that were used (HNO3, Triton X-100, dibasic ammonium phosphate) need to be provided. Additionally, information about the lead standard used should also be provided.
R: Thank you for the suggestion. In lines 94 - 102, we added section 2.2, “Chemicals and Reagents,” which describes the denomination of origin, manufacturers/suppliers, city, and country of the reactants.
- Similarly, the make and manufacturer of the instrument supplying the used deionized water should be provided. Also, the make and type of the water bath used should be provided.
R: In lines 95 – 102 and 123, we describe the make and manufacturer of the instrument supplying the used deionized water.
- Generally, appropriate information on the chemicals and instruments used should be provided.
R: This information was also included in the new section: 2.2 “Chemicals and Reagents”(lines 94 – 102.
Subsection 2.3
- Why did the authors conduct a duplicate analysis and not a triplicate analysis? Statistically, triplicate analysis is more robust than duplicate analysis. The authors should justify their reason.
R: In lines 125 - 135, we describe the validation test of our analysis, which was carried out following the recommendations of NOM-177-SSA1-2013 and the guidelines established by the Commission for Analytical Control and Expansion of Coverage (CCAYAC-CR-03/0). The repeatability test was performed, which consists of evaluating the lower limit of quantification by five-fold (2.5 µg/L, lower value of the calibration curve) and three concentration values located within the calibration curve: low level 8 µg/L, medium level 25 µg/L and high level 40 µg/L. For compliance with this test, the values obtained shall show a maximum coefficient of variation percentage of 20% for the limit of quantification and 15% for the rest of the determinations. The results obtained were 8.9%, 4.8%, 2.8%, and 0.8%, respectively. Therefore, the proposed method complies with the repeatability test. Consider compliance with the validation of the method decided to perform duplicate testing.
In results section
- Table 1 is not presented in a standard form suitable for journal publication. A 3-wier table format is more appropriate. The table is too long, boring and difficult to follow. The authors should decongest and summarize the table for easy readability.
R: We adjusted Table 1, only columns that were considered truly necessary to present the results were kept.
In discussion section
- The sentence in lines 186-188 is difficult to comprehend. This should be re-written.
R: We re-wrote the whole paragraph (lines: 182 - 185)
- I would suggest that sections 3 (results) and section 4 (discussion) should be merged as one section (results and discussion). This would make it easy for readers to link the concentration of Pb in this study to an explanation and comparison the authors intend with other studies. In this current form, it is difficult to go back and forth between the two sections to comprehend the authors' message, especially in the discussion section.
R: Thanks for the observation. The results and discussion section were left as a single section: “3. Results and discussion” and considering your comments we re-wrote the result and discussion section to have a more compressive lecture of this section.
- Again, the authors need to provide a concentration of any food product and not just its percentage contribution as it is in its current form. Providing the concentration detected in your discussion in comparison with other studies or MLs by FAO/WHO would give the readers an idea of the level of Pb contamination in that food sample.
R: According to the reviewer's suggestions, we added more studies to discuss the concentrations of Pb detected compared to our results (highlighted in yellow) throughout the text.
- “The authors need to be consistent with the use of scientific units. The author should stick to either concentration unit in ppb or mg/kg. Moreso, mg/kg is equivalent to concentration in ppm and not ppb. The authors need to be consistent. In lines 218-223, the authors referenced a study and reported in ppb.
R: All units were standardized to mg/kg for consistency (highlighted in yellow).
- The results were not critically and adequately discussed. Most of the results in this study were limited to other studies within Mexico. Putting it in the perspective of the global reader, the authors should make a more robust comparison with other studies within North/Latin America, and other continents. This would give the article a wider reach of audience, and room for global comparison.
R: We added evidence from other countries about Pb levels detected in different studies in order to discuss our results (highlighted in yellow) throughout the text.
- Most of the information in the latter part of the discussion is supposed to be in the introduction section. Also, there are too many repetitions of things already mentioned in the introduction section that are repeated in the discussion.
R: Thanks for this observation; we included more information in the discussion section and avoided repeating information.
Conclusion section
- The conclusion section did not mention anything about the results of this study and did not conclude on anything either.
R: Thanks to the reviewer's comment, the conclusion was rephrased (lines 363 - 370).
- The conclusion was poorly written. It needs to be completely rewritten.
R: The conclusion was rephrased according to the findings and comments (lines 363 - 370).
References
- A full bibliography of reference no 1 should be provided.
R: We reviewed all the reference section and added reference 1: “World Health Organization. 10 chemicals of public health concern [Internet]. Available from: https://www.who.int/news-room/photo-story/photo-story-detail/10-chemicals-of-public-health-concern

Reviewer 3 Report
Comments and Suggestions for Authors
your reference to QA/QC for this study needs to be added to. you need to show the QA/QC results for samples assessed during the analysis campaign carried out here. the degree of replication and repeat analysis. GFAAS is a sensitive technique - not only able to measure but also subject to wear and tear of furnace during large analysis programs.
would be nice to have some additional replication to show variability within food groups and a better comparison between this and other food safety surveys.
conclusions are very vague and don't provide any reference to new understanding. simply more policy... this is not a useful outcome from a detailed program of chemical analysis. Discussion should be about the significance of the results, implications for policy and whether ethical approval is needed and how information is disseminated.
Comments on the Quality of English Languagereasonable language. some small typographic errors
Author Response
Reviewer 3
- Your reference to QA/QC for this study needs to be added to. you need to show the QA/QC results for samples assessed during the analysis campaign carried out here. the degree of replication and repeat analysis. GFAAS is a sensitive technique - not only able to measure but also subject to wear and tear of furnace during large analysis programs.
R: Thanks for the reviewers' insightful comments. They improve our paper a lot. Authors share the concern that to address this, an internal standard product of acid digestion of a bovine liver mass (NIST) equivalent to 3.5 µg/L was placed each day of analysis. The percentage of recovery was calculated. The following information was added to the manuscript in section 2.3 Sample Analysis and Quality Control (lines 145 to 148).
- Would be nice to have some additional replication to show variability within food groups and a better comparison between this and other food safety surveys.
R: The authors agree with the reviewer. Unfortunately, the lack of monetary resources limits the number of food items and the number of samples for each food item. We consider that after this effort, authorities and other research groups would support more analysis. Information was added on the study's limitations in lines 342-350.
- Conclusions are very vague and don't provide any reference to new understanding. simply more policy... this is not a useful outcome from a detailed program of chemical analysis. Discussion should be about the significance of the results, implications for policy and whether ethical approval is needed and how information is disseminated.
R: We improve the conclusion section by considering the reviewer's recommendations (lines: 364-370).

Round 2
Reviewer 1 Report
Comments and Suggestions for Authors
This version of the work is significantly improved. The reagents used, procedures applied and quality control tests are introduced and described in detail. The paragraph about results has been expanded and the discussion is more complete. Although no methodological novelty has been developed, the work could be a useful starting point for future monitoring study based on more solid experimental design.
Regarding the Pb level detected in baby rice cereal (Brand 2), we corroborated this value, which is correct. …
I did not mean that the measured concentration in baby rice cereal (brand2) is wrong. I just suggested to analyse higher number of samples belonging to this group, since in reference 38, 168 samples were measured and the highest lead concentration was 67ppb. May be, baby rice cereal (brand2) it is not representative for the whole group baby food; in this sense it could be an outlier.
2)Pork Ham(brand1): the reported SD is wrong likely due typing mistake
R: Thanks for this observation. We have verified the information, and this data is correct.
This is a trivial question, anyway, in previously version of the work two replicates for this sample reported respectively 0.064 and 0.059 mg/Kg. The Average 0.062 is correct while the SD 0.017 is wrong (0.004 is correct!)
Reviewer 2 Report
Comments and Suggestions for Authors
I think the article has been substantially improved.
Comments on the Quality of English LanguageThe use of English Language has been improved.